Compacting oblivious agents on dynamic rings

http://orcid.org/0000-0003-4008-2445 Das Shantanu 1
Di Luna Giuseppe Antonio 2
Mazzei Daniele 3
Prencipe Giuseppe 3 giuseppe.prencipe@unipi.it
1 CNRS, LIS, Aix-Marseille University , Marseille , France
2 University of Roma "La Sapienza" , Rome , Italy
3 Dipartimento di Informatica, University of Pisa , Pisa , Italy
Galán José Manuel
Electronic publication date: 2021 Apr 22
Publication date: 2021
Volume: 7
Electronic Location ID: e466
Received 2020 Nov 11; Accepted 2021 Mar 12
Copyright: © 2021 Das et al.
Copyright year: 2021
Copyright holder: Das et al.
License: This is an open access article distributed under the terms of the Creative Commons Attribution License, which permits unrestricted use, distribution, reproduction and adaptation in any medium and for any purpose provided that it is properly attributed. For attribution, the original author(s), title, publication source (PeerJ Computer Science) and either DOI or URL of the article must be cited.
License URL: https://creativecommons.org/licenses/by/4.0/

Keywords: Dynamic networks, Mobile agents, Ring network, Compacting problem, Distributed computing

Funding: Università di Pisa, Pisa, Italy PRA 2018 43 This work was supported by Progetto PRA_2018_43 (Università di Pisa, Pisa, Italy). Giuseppe Di Luna was supported by the AXA Fellowship. There was no additional external funding received for this study. The funders had no role in study design, data collection and analysis, decision to publish, or preparation of the manuscript.

==============================
In this paper we investigate dynamic networks populated by autonomous mobile agents. Dynamic networks are networks whose topology can change continuously, at unpredictable locations and at unpredictable times. These changes are not considered to be faults, but rather an integral part of the nature of the system. The agents can autonomously move on the network, with the goal of solving cooperatively an assigned common task. Here, we focus on a specific network: the unoriented ring. More specifically, we study 1-interval connected dynamic rings (i.e., at any time, at most one of the edges might be missing). The agents move according to the widely used Look–Compute–Move life cycle, and can be homogenous (thus identical) or heterogenous (agents are assigned colors from a set of c > 1 colors). For identical agents, their goal is to form a compact segment, where agents occupy a continuous part of the ring and no two agents occupy the same node: we call this the Compact Configuration Problem. In the case of agents with colors, called the Colored Compact Configuration Problem, the goal is to group agents such that each group is formed by all agents having the same color, it occupies a continuous segment of the network, and groups of agents having different colors occupy distinct areas of the network. In this paper we determine the necessary conditions to solve both proposed problems. For all solvable cases, we provide algorithms for both the monochromatic and the colored version of the compact configuration problem. All our algorithms work even for the simplest model where agents have no persistent memory, no communication capabilities and do not agree on a common orientation within the network. To the best of our knowledge this is the first work on the compaction problem in a dynamic network.

Introduction

Research in the field of distributed computing has always considered fault tolerance as an important aspect of algorithm design and there are many studies on algorithms tolerating failures of nodes or links in a network. However, in recent years computing over a dynamic distributed system has become popular, mainly due to peer-to-peer systems, the intense distribution of mobile devices and the impact of sensors networks. In particular, in dynamic networks the system can experience topological changes that are not localized and sporadic; on the contrary, the topology changes continuously and at unpredictable locations, and these changes are not anomalies (e.g., faults) but rather an integral part of the nature of the system (Casteigts et al., 2012; Flocchini et al., 2006, 2008; Kuhn & Oshman, 2011). Dynamic networks model modern systems such as, for instance, wireless networks. In modern wireless networks nodes move continuously changing the induced communication graph. Moreover, thanks to the technological advancements introduced by Software Defined Networking (SDN) also the once static wired setting is acquiring a dynamic dimensions: routing paths and connections among node may frequently change orchestrated by the SDN controller.1

A general model for dynamic networks is the evolving graph model, where the dynamic network is modelled as a sequence of graphs, all having the same set of nodes, and where the set of edges can dynamically change over time; also, each graph in the sequence is a subgraph of the footprint graph which represents the overall underlying topology. In order to allow useful tasks to be performed on such a network, it is necessary to make few assumptions on the network connectivity: in particular, one common model assumes 1-interval connectivity: the network always stays connected, regardless of edges that might appear and disappear (see e.g., Kuhn & Oshman, 2011).

The study of distributed computations in these kinds of networks has mainly focused on problems related to information diffusion, reachability, agreement and several other communication problems (see e.g., Di Luna & Baldoni, 2015; Biely et al., 2015; Casteigts et al., 2014; Haeupler & Kuhn, 2012; Jadbabaie, Lin & Morse, 2003; Kuhn, Lynch & Oshman, 2010; Kuhn, Moses & Oshman, 2011; Ren & Beard, 2005). These studies adopt the message passing approach, under various different models of dynamic changes of topology.

An alternative way to deal with highly dynamic environments is to use mobile code: processes migrate from node to node of the network. Such processes are also known in the literature as mobile agents, where an agent is indeed an autonomous process that moves along the edges of the network and can perform computations at its nodes, using its own private memory and state information, as well as the information stored in each of the visited nodes.

In the last few years, several different models for mobile agents have been introduced, depending on their model of memory, of vision range, of communication and computation. In particular, there has been a lot of research on mobile agents moving in static networks. Here, the main studied problems have been exploration (Das, 2019) (a team of agents has to visit all nodes of the network) and patrolling (Kawamura & Kobayashi, 2015; Czyzowicz et al., 2017) (nodes have to be periodically visited). Work has also been done on coordination problems, where the agents are required to form a specific configuration. On this topic, one of the most studied problem is the rendezvous (Pelc, 2019) (or gathering), where all agents have to meet at a single node of the network. This problem has been studied both for agents with identities and anonymous (and thus identical), and for homonymous agents (i.e., where multiple agents share the same color or name). In the latter case, the problem of grouping the agents into teams with specific colors is called the team assembling problem (Liu et al., 2018).

The investigation on the use of mobile agents within dynamic graphs started relatively recently: following the way pursued in the static networks context, these studies focused mainly on the problems of exploration, patrolling and gathering (Gotoh et al., 2020; Mandal, Molla & Moses, 2020; Das, Di Luna & Gasieniec, 2019a; Di Luna et al., 2016, 2018; Ilcinkas, Klasing & Wade, 2014; Ilcinkas & Wade, 2013), all assuming the 1-interval connected networks. Under weaker models of connectivity, the only problem ever studied, to the best of our knowledge, is a weaker version of the gathering, where all agents but one gather (Bournat, Dubois & Petit, 2018). An up-to-date survey on computing by mobile agents on dynamic graph is in Di Luna (2019).

We finally note that the dynamicity can be either adaptive or not (Augustine, Pandurangan & Robinson, 2016). When adaptive, the sequence of graphs generated by the network dynamics depend by the choices made by the algorithm: more precisely, the scheduler deciding the dynamics of the network can inspect the state of the nodes to generate the worst possible scenario. In the non-adaptive case this sequence is decided apriori, before the algorithm starts. In the case of non-randomized algorithms, that is the one we consider in this paper, the two models are equivalent (the choices of a deterministic algorithm are predetermined).

Our Contribution. In the standard definition of the gathering problem, all agents (or all agents in the same team) in the end must be at the same node of the network. However, it might not be always physically possible for a single node to host a great number of agents at the same time. Motivated by this observation, in this paper we define and study the Compact Configuration Problem (CCP) and the Colored Compact Configuration Problem (ColoredCCP) problems: we have c teams of agents, where all agents in a team share the same color; in CCP c = 1, while in ColoredCCP c > 1. Initially the agents are scattered over a dynamic network G. The agents are required to move over the network so that, within finite time, all nodes of G occupied by agents having the same color induce a connected subgraph of G. In other words, we require the agents to group according to their colors, with all groups being disjoint, despite the chance of edges in G that might appear and disappear over time.

An important aspect of the proposed solution is that it works with agents that do not have memory of the past: in other words, the agents are oblivious. The importance of obliviousness comes from its link to self-stabilization and fault-tolerance (Dijkstra, 1982; Dolev, 2000); in addition to robustness, its practical advantage comes from the fact that it does not require any persistent memory (except for storing and executing the algorithm itself); its theoretical relevance derives from the fact that its presence renders the robots computationally weak and the solution to problems even more challenging. the research on the impact and limitations imposed by obliviousness has been investigated quite a lot in the literature (e.g., in Lamani, Potop-Butucaru & Tixeuil, 2010; Flocchini et al., 2013; Bérard et al., 2016; Ilcinkas, 2019).

To the best of our knowledge, even if loosely related to some problems studied in the context of autonomous mobile robots that can move on a plane (such as the near-gathering in Pagli, Prencipe & Viglietta (2015)) or the more recent (Bhagat et al., 2020), this is the first time this problems is studied in the context of a dynamic network populated by a distributed teams of autonomous and mobile agents. In this paper, to better understand the difficulties of the problem, we restrict ourselves to the ring network. In a ring, solving the CCP problem requires all agents (all of them have the same color) to occupy the nodes of a continuous segment of the network, with each node occupied by at most a single agent. With c > 1 teams (ColoredCCP), in the end all teams are required to occupy different sections of the ring.

Although conceptually simple, a ring is highly symmetrical, and it is quite often challenging to solve problems requiring symmetry breaking, like the ones studied here. We assume that neither the nodes nor the agents possess any unique identifiers, which makes the problem much harder. Moreover we consider the network to be dynamic: in particular, we assume the network to be 1-interval connected (at most one edge of the ring might be missing at any time). The results shown in this paper provide a full characterization of the solvable instances for both CCP and ColoredCCP. In particular, we show that only local visibility is not sufficient for solving the problem, even if the agents have unbounded memory. On the other hand, with global visibility of the network, even oblivious agents (i.e., agents with no persistent memory) can solve the problem.

The structure of the paper is as follows: in “Preliminaries” we formally define the problem; in “Basic Impossibilities” impossibility results are reported; in “CCP with Global Snapshot” we present the solution for CCP with Global Snapshot and c = 1; “ColoredCCP with Global Snapshot and c > 2” and “ColoredCCP with Global Snapshot and c = 2” introduce the protocols for ColoredCCP with Global Snapshot and c ≥ 2; “Conclusions” concludes the paper. Finally, a summary of all the results is reported in Tables 1 and 2.

Table 1 Summary of the results for the case of c = 1. Note that it is not possible to have a configuration that is asymmetric when h = 2 (this is indicated by the – in the table).

	Periodic	Mirror Symmetry	Asymmetric	
h = 2	Imposs. (Th. 2)	Imposs. (Th. 4)	–	
h > 2	Imposs. when the symmetry axis goes through two empty nodes. (Th. 6)	Solvable (Th. 5)	
Poss. when the symmetry axis goes through a node occupied by agents or an edge. (Th. 7)	

Table 2 Summary of the results for the case of c > 1.

	Periodic	Mirror Symmetry	Asymmetric	
c > 2	Imposs. (Th. 2)	Imposs. when the symmetry axis goes through either two empty nodes, or an empy node and one edge, or two edges and c > 3 (Th. 4)	Solvable (Th. 8 for h > 2) (Th 9 for h = 2)	
Imposs. when the axis of symmetry goes through either at least one occupied node, or two edges and c = 3. (Th. 11)	
c = 2	Imposs. when the symmetry axis goes through two empty nodes. (Th. 6)	Solvable (Th. 12)	
Poss. when the symmetry axis of symmetry does not go through two empty nodes (Th. 13)	

Preliminaries

We model a dynamic network as a graph where edges can change over time. The changes are decided by a fictional omniscient adversarial entity. On top of this dynamic graph a set of agents move, along the edges of the graph, with the final goal of forming a compact segment. In the following, we introduce the main definitions used throughout the paper. The system is synchronous: agents perform their operations in discrete time units called round. Rounds are univocally mapped to numbers in N, starting from 0. All agents start the execution at round 0.

Interval Connected Ring. A dynamic graph G is an infinite sequence of static graphs (G0,G1,…).

For each round r∈N we have a graph Gr:(V, E(r)) where V:{v0,…,vn − 1} is a set of nodes and E:N→2(V2) is a function mapping a round r to a set of undirected edges. Given a dynamic graph G, its footprint G is the graph obtained by the union of all graph instances G=(V,E∞)=(V,∪i=0+∞E(i)). A dynamic graph G is a 1-interval connected ring if its footprint is a ring and Gr is connected, for each round r. In this paper, we assume 1-interval connected ring such that at most one edge of the ring can be missing at any time; such an edge is arbitrarily chosen by an adversary. Throughout the paper we will refer to such a network by dynamic ring. The graph G is anonymous, i.e., all nodes are identical to the agents, the endpoints of each edge are unlabelled, and we do not assume any common orientation (i.e., the ring is not oriented).

The agents. We consider a set of autonomous agents, A = {a1,…,ak} that are initially located on distinct nodes of a dynamic ring. Each agent has an initial color in [0, c − 1] (when c = 1, all agents have the same color). When c > 1, we assume that the sets of agents having the same color all have the same size h, with h ≥ 2 , that the size of the ring is at least 2hc + c, and that there exists a total ordering on the colors; in particular, we call first_color the first color in this ordering. Also, the color assigned to each agent is fixed at the beginning and it cannot be changed.

All agents execute a sequence of Look, Compute, Move cycles. In our (synchronous) system, each Look, Compute, Move cycle is executed at the beginning of each round, and it takes exactly one round to complete. In the Look phase of each cycle, the agent gets a snapshot of the environment. In the Compute phase the agent uses the information obtained from this snapshot to compute the next destination, which may be the current node or one of its neighbours; all agents run the same algorithm. Finally, during the Move phase an agent traverses an edge to reach the destination node. Given a direction of movement, we say that an agent a is blocked by the missing edge if the edge adjacent to a, in the chosen direction of movement, is missing. Note that blocked here refers only to the fact that the current direction of the agent is blocked by a missing edge; thus, it does not imply that the agent cannot change direction, hence follow an edge that is indeed alive.

We say that two agents collide if they occupy the same node at the same round. When two (or more) agents with distinct colors occupy the same node, we say that the collision is admissible.

The agents are oblivious, that is they have no persistent memory. This means that the robots have no memory of past actions and computations, and the computation is based solely on what determined in the Look of the current cycle. In other words: for the robots, every configuration occurs as if for the first time.

The visibility of the agents may be either global or local:Global Snapshot: The snapshot obtained by an agent in round r contains the graph Gr (with the current location of the agent marked), and, ∀v ∈ Gr, the colors of the agents (if any) that are located in node v.

Local Snapshot: The snapshot obtained by an agent placed at a node v in round r contains the same information as in the Global Snapshot model for all nodes at distance at most R from v.

Configurations and other definitions. The configuration of the set of agents A at round r is a function Cr:A→V that maps agents in A to nodes of V where agents are located. The initial configuration is the configuration of agents at round 0; when clear by the context, we will use C to denote the current configuration. We denote by Cr(A) the set of nodes where agents in A are located at round r, and by G[Cr(A)] the subgraph induced by the locations of agents in A in graph G at round r.

A segment is a set of nodes of G that have connected footprint and that do not form a cycle. Given a node v ∈ G we say that the node is empty at round r if in Cr there is no agent on v. Similarly, we say that a segment of nodes is empty at round r if all nodes of the segment are empty. We say that a segment is full if each node of the segment contain agents of the same color.

A full segment S is blocked by the missing edge if an agent in S is blocked according to its chosen direction of movement. Also, we say that S moves when all agents in the segment do a move in a given direction. Given two disjoint segments, the distance between them is the minimum number of nodes between two endpoints of the segments.

Finally, the configuration of a set of agents is said to be (refer to Fig. 1): (1) Periodic if the agents are placed periodically on the ring, that is there is a rotational symmetry of less than 2π; (2) Mirror Symmetric if the configuration contains an unique axis of symmetry; i.e., a rotational symmetry of 2π (in this case, we will also say that the configuration has a Mirror Symmetry); and (3) Asymmetric if the configuration is neither periodic nor mirror symmetric.

Figure 1 Examples of configurations that are: (A) periodic with 4 axes of symmetry; (B) with a mirror symmetry, and (C) aperiodic.

The Compact Configuration Problem. We are now ready to introduce the two problems that will be investigated in the remainder of the paper. In the CCP problem the agents, initially arbitrarily placed, move to form one full segment (i.e., with no empty nodes).

Definition 1 (Compact Configuration Problem). Given a dynamic graph G with footprint G and a set of agents A, we say that an algorithm solves the distributed Compact Configuration Problem (CCP) if and only if there exists a round r, when G[Cr(A)] is connected and each agent occupies a distinct node.

For multi-colored agents, our goal is for agents of the same color to occupy continuous segments, while agents of distinct colors are separated. Interestingly, if agents of different colors cannot occupy the same node at the same time, then it is impossible to form two disjoint full segments. We will show this fact by constructing a counter-example, as described by the following:

Theorem 1. Given a dynamic ring and two coloured set of agents of size 4, there exists no algorithm that, from any possible starting configuration, is able to form two non-overlapping full segments, while avoiding collisions of agents having different color.

Proof. Starting from the configuration in Fig. 2A, we explored the space of all possible solutions using a computer-assisted method. We define a state to be a binary string of 8 digits, with exactly four digits being equal to 0 (the first color) and other four equal to 1 (the second color), and such that the string is not sorted (neither increasing nor decreasing). The state represents a configurations in which agents having two different colors are interleaved. For instance, the state of the configuration in Fig. 2A is 00010111.

Figure 2 Example of (A) a configuration where solving ColoredCCP with only swap of agents is not possible, and (B) a configuration that is a solution for our definition of ColoredCCP.

In particular, we examined 54 different possible states; actually, the number of possible different states is larger: however, we restricted the space of states by grouping both complement configurations (i.e., 00100111 and 11011000) and cyclically shifted configurations (i.e., 00010111 and 10001011, that are cyclically shifted by 1 position to the right). For each of the examined states, we verified that an adversarial scheduler is always able to block an edge such that, for any possible switch of agents, it is not possible to reach a configuration with two non-overlapping full segments (i.e., state 00001111).

The code used to explore the space of possible solutions can be accessed at the following url: https://colab.research.google.com/drive/1W1H27vdTLC3cEs2rYc2k8TO3ivbppjOR.

Unfortunately, we do not know whether the previous impossibility holds also when then number of agents with the same color is 3.

Because of the result proven by previous theorem, in the ColoredCCP problem, we require all agents having the same color to form one full segment, and that at most two of these full segments intersect (see Fig. 2B).

Definition 2 (Colored Compact Configuration Problem) Given a dynamic graph G with footprint G and sets of agents Ai, where all agents in the i-th set have the same color i, with i ∈#x2208; [1,c] and c ≥ 2, we say that an algorithm solves the distributed Colored Compact Configuration Problem (ColoredCCP) if and only if there exists a round r where, for each i ∈ [1,c]: (i) each agent in Ai occupies a different node and G[Cr(Ai)] is connected; and (ii) there exists at most two distinct colors p and j such that G[Cr(Ap)] and G[Cr(Aj)] intersect.

In the following, we will refer to a configuration that satisfies either Definition 1 or Definition 2 as a compacted configuration.

Basic Impossibilities

In this section, we will show under which conditions the problem is not solvable.

Periodic configurations

Let us start with a general results, that holds also in case the ring is not dynamic.

Theorem 2 Given a ring G, and a set of agents A initially placed on G0 in a configuration that is periodic and not compacted, it is impossible to solve the CCP or the ColoredCCP problem, even in the Global Snapshot model.

Proof. In a periodic configuration, the ring can be partitioned into identical non-full segments. In case no edge is ever missing, the initial symmetry between the agents cannot be broken deterministically: in fact, agents occupying equivalent positions in different segments can only take the same action in each step; thus, the configuration can only stay periodic. Finally, by observing that any compacted configuration (with k < n) is not periodic, the theorem follows.

Therefore, in the following we will assume that the initial configuration is either asymmetric or contains a mirror symmetry.

Local snapshots

In this section we show that the compaction problem cannot be solved in the Local Snapshot model, even when the initial configuration is asymmetric. The visibility graph of a configuration C is defined as the graph Gvis = (A, E), where A is the set of agents and (a, b) ∈ E whenever agent b is within distance R from a.

Theorem 3 In the Local Snapshot model, starting from a configuration C such that C is asymmetric and has a connected visibility graph, there is no algorithm that solves CCP, avoiding collisions, even if the agents have unbounded memory.

Proof. The proof is based on the concept of local view of an agent: the local view is the part of the entire configuration that the agent can see. More formally, given an agent a at node v of G with visibility radius R, its local view is an ordered list of 2R elements; the element of the list in position j is of the form “distance j:(L:vL, R:vR)”, where vL (resp., vR) is either 0 or 1 depending on whether the j-th node to the left (resp., to the right) of a is empty or not (being the ring unoriented, left and right refer to the local notion that a has of left/right). For instance, let us consider the agent a1 in Fig. 3 with visibility radius 3: its local view is [distance 1: (L:0, R:1), distance 2: (L:1, R:1), distance 3: (L:1, R:1)].

Figure 3 An asymmetric configuration where CCP is unsolvable with visibility radius R = 3.

Let us now consider the configuration C depicted in Fig. 3, with R = 3, and let us first focus on agents a3 and a4: they both have the same local view [distance 1: (L:0, R:1), distance 2: (L:1, R:1), distance 3: (L:1, R:1)] (note that being the ring unoriented, they may have different notion of left/right and clockwise/counter-clockwise direction). Therefore, they either both move or they both stay still. In case they both move, there will be a collision. Thus, to avoid collision, they should not move. The same holds for the pair of agents a5 and a6, that also have the same view of a3 and a4, and for the pair of agents a2, a7. In contrast, a1 and a8 have a different view: [distance 1: (L:0, R:1), distance 2:(L:1, R:1), distance 3: (L:1, R:1)].

Let us now focus on agent a1: before deciding to move to v2, it has to be sure that a2 does not decide to do the same. Therefore, a1 tries to infer the local view of agent a2. In particular:a1 sees that a2 has an empty node (i.e., v2) and an occupied node (i.e., v3) at distance 1;

a1 sees that a2 has an occupied node at distance 2 (i.e., a1), but cannot see whether the other node at distance 2 from a2 (i.e., v4) is occupied or not; and

a1 sees that a2 has an occupied node at distance 3 (i.e., v1), but cannot see whether the other node at distance 3 from a2 (i.e., v5) is occupied or not.

Therefore, a1 can only infer a partial view of a2 (i.e., [distance 1: (0, 1), distance 2: (1, ·), distance 3: (1, ·)]); and it cannot decide whether a2 has a different view from its own view (notice that a1 is not aware of the orientation of a2). Hence, it cannot decide to safely move to v2 without the risk of colliding with a2.

The same argument also holds for a2, a7 and a8; thus, agents a1, a2, …, a8 cannot move. Therefore, none of the agents can move if they want to avoid collision. Hence it is not possible to reach a configuration in which agents form compact lines. The same argument can be extended for agents having any visibility radius R by using a sufficiently large ring.

We note that the previous theorem holds for any ring, even non dynamic ones.

The case of two agents

Finally, in this section we examine the very special case of having only two agents in the system. It is clear that in such a case only the monocromatic version of the problem makes sense: in fact, if the two agents have different colors, then the problem is solved by definition. Surprisingly, solving CCP with two agents, in arbitrary initial configurations, is impossible.

Theorem 4 Let us consider an arbitrary dynamic ring G and an arbitrary initial configuration with two agents. Then, it is impossible to solve CCP.

Proof. First, notice that if the two agents are antipodal, the configuration is periodic and, by Theorem 2, the theorem follows. Thus, let us assume that at the beginning the two agents are not antipodal.

Also, notice that with only two non-neighbors agents in the ring, the configuration has always an unique axis of symmetry, say ax. If ax, passes through two empty nodes, the problem cannot be solved: in fact, any movement of the agents would keep ax passing through two empty nodes (remember that a collision of agents with the same color is not admissible), the two agents can never become neighbors, and the theorem follows.

Hence, ax has to pass through at least one edge. Note that, the only possible strategy for the agents to form a full segment is to move towards one of the two edges crossed by ax, say e. Referring to the example depicted in Fig. 4, we distinguish the two possible cases:

ax passes through a node (Fig. 4A). As stressed before, to solve the problem the agents can only try to reach e. If, during this movements, one of the two agents cannot move because of a missing edge (Fig. 4B), either they stay still forever (and CCP cannot be solved), or one of them moves. In this second case, the configuration stays as a new axis of symmetry passing through an empty node and edge e′ that is antipodal with respect to e (Fig. 4C). In order to correctly achieve compaction, the agents now have to start converging towards e′. If, during these movements, one of the edge is missing, this argument can be iterated, hence a compacted configuration never achieved, and the theorem follows.

ax passes through two edges (Fig. 4D). Let e the edge elected by the agents (being the configuration aperiodic, agents can always elect e). If, during these movements, one of the two agents cannot move because of a missing edge (Fig. 4E), either they stay still forever (and compaction never achieved), or one of them keeps moving in the same direction: in this second case, the configuration has a new axis of symmetry passing now through two empty nodes (Fig. 4F). Now, by previous Case 1, CCP cannot be solved. Therefore, the other option they have is to switch direction, and start moving towards the edge e′ on the axis of symmetry, antipodal to e (Fig. 4G). In this case (Fig. 4H), we end up again in a scenario similar to the one in Fig. 4D. Hence, by iterating the argument, we can conclude that a compacted configuration is never achieved, and the theorem follows.

Figure 4 Proof of Theorem 4. (A) Axis passes through one edge and one node. Agents move towards the elected edge. (B) One of the agents is blocked, the other moves. (C) The symmetry axis changes, as well as the elected edge. (D) Axis passes through two edges: agents move towards the elected edge. (E) One of the agents is blocked. (F) If the other agent moves, a configuration where CCP is unsolvable is reached: the axis passes through two nodes. (G) The agents have to move in the other direction. (H) The configuration is symmetric to the initial one (i.e., the configuration in (D)).

CCP with Global Snapshot

Because of the impossibility results stated in the previous section, in the following we will consider the Global Snapshot model. Furthermore, we will also assume that the initial configuration is aperiodic (i.e., it is either asymmetric or with a mirror symmetry), and that there are more than two agents in the system.

The asymmetric case

First, let us consider the case when the initial configuration is asymmetric. Let Er be the empty segment of maximum size in the configuration at round r. If, at round r = 0, there is more than one empty segment of maximum size, we can deterministically elect one of these (since the initial configuration is asymmetric).

Let S1 and S2 be the maximal full segments of length at least 1 on the two sides of segment Er (see Fig. 5A). In case |S1| ≠ |S2|, without loss of generality let |S1| < |S2|; we define the augmented S1, denoted by S + 1, as the block of nodes constituted by the nodes in S1 (all non empty), plus the empty node v close to S1 and not in Er, plus, if any, all agents between v and the next empty node (moving away from S1, see Figs. 5B and 6A).

Figure 5 Asymmetric initial configuration. (A) |S1| = |S2|. (B) |S1| < |S2|.

Figure 6 (A) Definition of S1+. (B) Movement of S1+ (the arrows denotes the direction of movement).

The algorithm for solving CCP tries to increase the length of the empty segment Er in each step, while preserving the asymmetric configuration. This is done by moving either S1 or S2 or both. The details are reported in Algorithm 1.

Algorithm 1 One Color Connected Formation.

Pre-condition: Initial configuration is asymmetric.	
Let Er be the empty segment of maximum size in the current configuration. If there is more than one empty segment of maximum size, we can deterministically select one of these as segment Er (since the initial configuration is asymmetric).	
Let S1 and S2 be the non-empty maximal segments adjacent to the chosen empty segment Er. Let a1 and a2 be the agents closest to S1 and S2 respectively (going away from Er).	
1. If the smallest distance between S1 and S2 is strictly greater than one:	
(a) If |S1|=|S2|,	
• If neither S1 nor S2 is blocked, they both move away from Er.	
• Otherwise, let di be the distance between Si and ai,	
– If d1=d2, the segment that is not blocked moves away from Er.	
– Otherwise, without loss of generality, let d1<d2.	
* If S1 is not blocked, then S1 moves away from Er.	
* If S1 is blocked, then all agents not in S1 move towards S1 (preserving the distance d2).	
(b) If |S1|≠|S2|, without loss of generality, let |S1|<|S2| (refer to Fig. 5B). S1+ and S2 move away from Er.	
2. Else: let v the only empty node separating S1 and S2. If the largest among the segments S1 and S2 is not blocked, this segment moves towards empty node v. Otherwise the other segment moves towards node v.	

Lemma 1 Starting from an asymmetric configuration, by executing Algorithm One Color Connected Formation , at any round r ≥ 0:|ℰr| > |ℰr−1|, and

The configuration is either asymmetric or solves CCP.

Proof. Let S1 and S2 be as defined in Algorithm 1. We proceed by induction on the number of rounds. By the precondition, the starting configuration is asymmetric. Let us now assume that the inductive hypothesis is true for round r −1. Let us consider the possible cases of the algorithm starting from the configuration at the beginning of round r:1. If the smallest distance between S1 and S2 is strictly greater than one, by construction, we have the following cases (refer to Fig. 5):

If |S1| = |S2|, and neither S1 or S2 is blocked, they both move away from Er, and by induction both (i) and (ii) hold. What happens is that both distances d1 and d2 decrease by one. If one of the distances reaches value 0, then the respective full segment Si increases, and the asymmetry is kept. If both distances go to 0, both segment increase, and the asymmetry is kept by induction hypothesis.

If |S1| = |S2|, S1 is blocked (the case when S2 is blocked is symmetric), and d1 = d2, then S2 moves away from Er, one of the distances became different from the other, thus introducing a new asymmetry. Moreover, apart from Er, no other empty segment increases its size. Therefore, both (i) and (ii) hold.

If |S1| = |S2|, S1 is blocked, and d1 < d2, then all agents not in S1 move towards S1 away from Er. Two cases may occur: if d1 does not reach 0, then the asymmetry is kept (since d2 does not change); otherwise, if d1 becomes 0, we have that |S1| ≠ |S2|, i.e., the configuration stays asymmetric. Therefore, both (i) and (ii) hold.

If |S1| = |S2|, S1 is not blocked, and d1 < d2, then S1 moves away from Er . By using the same arguments of the previous case, it follows that both (i) and (ii) hold.

If |S1| < |S2| then, S1+ and S2 move away from Er. Thus, |Er| > |Er−1|. If before the movement d1 >1 then after the movement the asymmetry is kept, since |S1| ≠ |S2|. Otherwise, d1 = 1 before and after the movement, hence the asymmetry is preserved. Thus, both (i) and (ii) hold.

2. If the smallest distance between S1 and S2 is exactly one (see Fig. 7), let v the only empty node separating S1 and S2. Since by inductive hypothesis the configuration is asymmetric, we have |S1| ≠ |S2|. Also, let e1 and e2 be the edges between S1 and v, and between S2 and v, respectively. Since at most one between e1 and e2 can be missing, by the algorithm, one full segment is formed within one round, and the lemma follows.

Figure 7 Case 2 of Algorithm 1: the distance between S1 and S2 is 1.

In all cases, the lemma follows.

By previous lemma, since the size of Er strictly increases at each round, we can state the following:

Theorem 5. If the initial configuration is asymmetric, the agents executing Algorithm ONE COLOR CONNECTED FORMATION , solve CCP within at most n rounds.

The case of mirror symmetry

Let us now consider the case where in the initial configuration C there exists an unique axis of symmetry. Note that if there are two or more axes of symmetry, then the configuration is periodic.

Theorem 6. Let the initial configuration be aperiodic with an unique axis of symmetry, and not compact. Then, if the axis of symmetry passes through two empty nodes, then CCP is not solvable.

Proof. Let us assume that the problem is solvable, and that, by contradiction, the axis of symmetry of the initial configuration passes through two empty nodes (see Fig. 8). If no edge is missing during the algorithm, the agents in both sides of the axis perform symmetric actions and the configuration stays with the same axis of symmetry. Since the agents avoid collision, no agent can move to the nodes on the axis; therefore, CCP cannot be solved in this case.

Figure 8 Example configurations for CCP in the case of a single axis of symmetry.

In Algorithm 2, we present a solution for CCP with more than 2 agents, when the initial configuration is aperiodic and the axis of symmetry either (a) passes through at least one edge and it does not pass through a non empty node, or (b) passes through at least one non empty node. By Algorithm 2, and by Theorem 5, we can state the following:

Algorithm 2 One Color Mirror Symmetry.

Pre-condition: Initial configuration is aperiodic with an unique axis of symmetry, with more than two agents. The axis of symmetry does not pass through two empty nodes.	
(a) If the axis of symmetry passes through at least one edge. Since the configuration is aperiodic, we can elect a unique edge e that is crossed by the axis of symmetry. Once e has been elected, the two agents nearest to e that do not belong to a full segment containing e, are selected to move towards e. If none of these agents are blocked by a missing edge, the symmetry axis is preserved after the moves of the agents. Otherwise, if an agent cannot move because of a missing edge, the next configuration becomes asymmetric, and Algorithm 1 can be applied.	
(b) If the axis of symmetry passes through at least one non empty node. In aperiodic configurations, it is always possible to elect one of the agents (agent a) among those that occupy the nodes crossed by the unique axis of symmetry.	
1. If the neighbor nodes of a are empty, a moves to one of the neighbors (chosen arbitrarily when both incident edges are available); After the move, the configuration becomes asymmetric and Algorithm 1 can be applied.	
2. If the two neighbor nodes of a are both occupied, and the axis of symmetry passes through another node occupied by agent b, and the two neighbor nodes of b are both empty, then b moves to one of the neighbors (chosen arbitrarily when both incident edges are available); After the move, the configuration becomes asymmetric.	
3. If no agent on the symmetry axis can move, since the configuration has a mirror symmetry, there must be two (full) maximal segments of equal size to both the left and the right of a. These two segments move away from a by one position. Now, either the configuration becomes asymmetric (if one of the two segments cannot move because of a missing edge), or previous Case b.1 applies.	

Theorem 7 If the initial configuration has an unique axis of symmetry, more than two agents, and the axis of symmetry either (a) passes through at least one edge and it does not pass through a non empty node, or (b) passes through at least one non empty node, then CCP is solvable.

Proof. We distinguish the two possible cases:The axis of symmetry passes through at least one edge. Let a and b be the two agents that do not belong to the full segment S containing e. Note that a and b have to exist, otherwise the problem is solved. Also, if only one of them exists, then the configuration cannot be symmetric with the axis of symmetry passing through e, contradicting the assumption. These two agents move towards e using two distinct edges, let them be ea and eb. Now, two scenarios may occur. (i) Edges ea and eb are both alive: in this case the distance between a and b and S decreases; when this distance becomes 0, segment S increases. It is clear, that if this scenario always applies all agents eventually join S. (ii) Either ea or eb is missing: in this case only one among a and b moves leading to an asymmetric configuration. In this case, Algorithm 1 can be applied and, by Theorem 5, the theorem follows.

The axis of symmetry passes through at least one occupied node. Let us analyse the three possible cases of Algorithm 2(b).In this case a moves leading to an asymmetric configuration (note that the adversary cannot prevent a to move since it can block at most one edge). In this configuration, Algorithm 1 can be applied, and by Theorem 5 the theorem follows.

The proof in this case is similar to the proof of the previous one.

Let a be the agent and S− and S+ be the two full maximal segments to the left and to the right of a. Both of them try to move away from a, and at most one of them can be blocked by an adversary. If none of them is blocked, then we reach a configuration in which both neighbour locations of a are empty, and thus the previous case applies. If only one can move, an asymmetric configuration is reached. In this configuration, Algorithm 1 can be applied and, by Theorem 5, the theorem follows.

ColoredCCP with Global Snapshot and c > 2

In this section, we investigate the compaction problem for heterogenous agents having c > 2 distinct colors; recall that h is the number of agents of each color. The problem is trivial when h = 1.

Asymmetric initial configuration and h ≥ 3

The algorithm for this case builds segments around some specific points of the ring, called rally points. These points are identified during the execution of the algorithm, and to each color is assigned a specific rally point.

Definition 3. We say that agents are forming a compact line if they are forming a full segment of size h around the rally point of their color. We say that agents are forming an almost compact line if they are forming a full segment of size h − 1 around the rally point of their color; the only agent that is not part of the almost compact line is called a dangling agent.

Moreover, let FC denote the set of agents colored with first_color. We say that the current configuration is correctly placed if and only if both the following conditions hold on all the colors different from first_color:(i) There are at least c − 2 compact lines that do not overlap;

(ii) There is at most one almost compact line.

The MULTI COLOR CONNECTED SEGMENT algorithm is split into three main steps, described in Algorithm 1, 4, and 6, respectively. Let us first describe the intuition behind each step.

First Step (Algorithm 3). The main idea of the first step is to make an agent with color first_color move in such a way that all agents with color first_color become asymmetrically placed (this step is skipped if FC is already asymmetric). Once FC is asymmetric, the agents in FC do not move until the last phase of the algorithm: these agents are used as reference points to univocally identify both the rally points and a unique orientation of the ring.

Second Step (Algorithm 4). In the second step, the algorithm proceeds by making each color but first_color to form a full segment around the respective rally point; that is, after this step the configuration becomes correctly placed.

Third Step (Algorithm 6). Once the configuration is correctly placed, the only agents still to fix in order to solve the problem, are the agents in FC (that are still asymmetrically placed), and the (at most one) dangling agent (this agent has a color different from first_color). Note that, if there is no dangling agent, then there are c − 1 compact lines, and no almost compact line.

Algorithm 3 Multi Color Connected Segment (First Step).

Pre-condition: Current configuration is not correctly placed and FC is symmetric.	
Let a be the first agent in FC that is able to move, according to the total ordering: a moves one step to make FC asymmetric.	

Algorithm 4 Multi Color Connected Segment (Second Step).

Precondition: Current configuration is not correctly placed, and FC is asymmetric.	
During this step, FC never moves until current configuration is correctly placed. Since FC is asymmetric, it can be used to establish an orientation of the ring and a total order among the agents in FC. That is, each agent in FC can unambiguously assume an unique rank in [0,FC−1], and this rank can be computed by all agents in the system. Let vf be the node where the first agent in FC is located (i.e., the node where the agent with rank 0 is located).	
1. Rally Points Computation. FC is now used to compute c rally points, as follows: vf is the first rally point, rp0. The i−th rally point rpi is the node of the ring at distance i∗(2⋅h+1) from rp0 (in the clockwise direction; we assume the ring size is at least 2⋅h⋅c+c).	
2. Formation using Rally Points. The rally points are now used by all agents not in FC to form rally lines (Definition 5.2), by executing routine RALLY POINTS CONNECTED FORMATION (Algorithm 5).	

The idea here is to use the compact lines formed so far to establish a global chirality of the ring, and a rally point for FC. In particular, the already formed compact lines do not move, hence the computed chirality can be kept; the other agents (i.e., those in FC and the dangling agent) move following the same strategy used in the second step. The movements go on until either ColoredCCP is solved, or there are c − 1 compact lines and one almost compact line. In the latter case, the only dangling agent and the almost compact line (by construction, all these agents have the same color) move one towards each other until they form a compact line.

Since the initial configuration is asymmetric, we have the following:

Lemma 2 If in the initial configuration FC is not asymmetric, by executing Algorithm 3, within finite time agents in FC are placed asymmetrically on the ring.

Proof. The lemma follows by observing that there are at least two edges connecting agents in FC to nodes not occupied by any agent in FC, hence agent a can always be uniquely identified.

Once the agents in FC occupy asymmetric positions on the ring, it is possible to elect one of them as a leader, which provides a global orientation to the ring. Once a global orientation has been computed, the positions of agents in FC allow also to compute the rally points where all other agents will form their respective compact lines, as detailed in Algorithms 4. Let us denote these points by rpi, 0 ≤ i ≤ c − 1. A color i is assigned to each rally point rpi, 0 ≤ i ≤ c − 1, with color 0 = first_color assigned to FC.

Definition 4 Given a rally point rpi, let us call the rally line of color i a maximally full segment of color i that is formed around rpi. Extending Definition 3, we will call dangling any agent that is not part of a rally line.

ROUTINE RALLY POINTS CONNECTED FORMATION (Algorithm 5) makes all agents of color i gather around rpi.

Algorithm 5 Rally Points Connected Formation (Auxiliary routine).

There are c rally points, sorted according to the ring orientation. One of the following two patterns of movements will be executed, according to the verified preconditions.	
Pattern 1. There exists a rally line rli of color different from first_color that is being formed around rally point rpi that has at least two dangling agents. Let a be any of these dangling agents, and p be the counter-clockwise path that connects a with its own rally line.	
Movement (Fig. 9):	
• If a is not the farthest agent from its rally line (according to the counter-clockwise oreintation), and on p there is a missing edge, then a does not move.	
• If on p there is a missing edge, and a is the farthest agent from its rally line (according to the counter-clockwise direction), then a moves clockwise.	
• If on p there is no missing edge, then a moves counterclockwise.	
Pattern 2. For all rally lines of color different from first_color, there is at most one dangling agent; let m be the number of rally lines with exactly h−1 agents (i.e., only one dangling agent). Given a dangling agent a, let p be the counter-clockwise path that connects a with its own rally line.	
Movement (see Fig. 10):	
• If there are m−1 dangling agents that are blocked in the counter-clockwise orientation by a missing edge, a has the shortest clockwise distance to its own rally line among all clockwise distances of all other dangling agents from their own rally lines, and p has a missing edge, then a moves clockwise.	
• If the first edge on p is not missing, then a moves counter-clockwise.	

Figure 9 Pattern 1 of Algorithm 5. The bold node represents the rally point for agents having red color. (A) The dangling agents are not blocked. They move counter-clockwise towards their rally line. (B) The dangling agents are blocked. The last agent changes direction and moves clockwise towards its rally line.

Figure 10 Pattern 2 of Algorithm 5. (A) The black agent switches direction. (B) The vertical striped agent switches direction.

Lemma 3 Within finite time, by executing Routine RALLY POINTS CONNECTED FORMATION (Algorithm 5), the system reaches a configuration with c − 1 almost compact or compact rally lines.

Proof. If c − 1 rally lines are almost compact, the lemma trivially follows. Thus, let us assume that there exists at least one rally line, say rli, that has at least two dangling agents. By construction, only Pattern 1 of RALLY POINTS CONNECTED FORMATION can be executed. Let us consider only agents having color i. Let a be the closest agent in the counter-clockwise direction to rpi that has not reached rli yet. We will show that, within finite time, the size of rli increases. Note that, as long as there is no missing edge between a and rpi, a will always move towards its own rally line, even if other agents are blocked.

Therefore, if no edge on the path between a and rpi is ever missing, within finite time the size of rli increases by one unit and the statement trivially follows. Otherwise, let a′ be the furthest agent from rli: according to Pattern 1, a′ switches direction and starts moving clockwise towards rli. As long as a is blocked by a missing edge on its path towards rpi, a′ keeps approaching rli. If a′ becomes blocked before reaching rli, then a can perform at least one step (counter-clockwise) towards rli, thus decreasing its distance from rli. Thus, by iterating the above argument, within finite time either a or a′ will join rli.

In conclusion, within finite time, rli becomes almost compact, and the lemma follows.

Lemma 4 Let us assume that in the current configuration there exist m > 2 almost compact rally lines, and c − 1 − m compact lines. Within finite time, by executing Routine RALLY POINTS CONNECTED FORMATION in Algorithm 5, m decreases.

Proof. By construction, only Pattern 2 of RALLY POINTS CONNECTED FORMATION can be executed. Note that the agents that are already part of a rally line do not move any more. According to Pattern 2, each dangling agent moves towards its rally line, according to the counter-clockwise direction. First, let us assume that either m − 1 or m dangling agents are blocked by a missing edge (towards their way to their rally lines), and let a be the dangling agent that is closest to its rally line, according to the clockwise direction, and p be the counter-clockwise path that connects a with its own rally line. We distinguish the three possible cases:If m − 1 agents are blocked by a missing edge (on their counter-clockwise direction), and a is not one of them, and the missing edge is on p, then a moves clockwise towards its rally line. If a reaches its rally line, the lemma follows. Otherwise, when a becomes blocked (during its clockwise movements), the other m − 1 agents cannot be blocked anymore according to the counter-clockwise orientation, hence they can get closer (counter-clockwise) of at least one unit to their respective rally lines. Note that as long as one of the m − 1 agents does not reach its line, a will be the agent that is closer, clockwise, to its own rally line. By iterating this argument, within finite time m decreases, and the lemma follows.

If m agents are blocked, then a is one of them: in this case, a moves of one step clockwise. Therefore, either a joins its line, and the lemma follows, or previous case applies.

If m − 1 agents are blocked by a missing edge (on their counter-clockwise direction), and a is one of them, by construction there exists an agent, say b, that is not blocked, and that is moving counter-clockwise towards the m − 1 blocked agents. Within finite time, either b reaches its own rally line, or b reaches the the blocked edge, or the m − 1 agents become unblocked. In the first case, the lemma follows. In the second case, previous case applies. Otherwise, the m − 1 agents get closer to their rally lines. Thus, by iterating the argument, the lemma follows.

Now, let us assume that at most m − 2 dangling agents are blocked. One of the following holds: (1) one agent reaches its own rally line, thus m decreases and the lemma follows; (2) another agent will join the blocked ones, hence there will be either m − 1 or m blocked agents, and previous case applies.

Thus, by previous Lemmas 3 and 4, the following holds:

Lemma 5 Within finite time, by executing Algorithm 4, the configuration becomes correctly placed.

Finally, by executing Algorithm 6, agents are able to solve the problem. In particular, at the beginning of this step, there are at least c − 2 compact lines, at most one line with just one dangling agent, and the agents in FC that still needs to be compacted.

Algorithm 6 Multi Color Connected Segment (Third Step).

Precondition: Current configuration is correctly placed. Let da be the dangling agent of the almost compact line, if any.	
• Since agents in FC have to move, it is possible that the orientation of the ring that FC is establishing gets lost. Therefore, before moving any agent in FC, the other c−1 classes (one class per color) are used to establish a new orientation of the ring: in particular, let L2 and L3 be the set of agents colored with the second and the third color in the total ordering. The agents in L2 and L3 are either both already compacted, or (at most) one of them forms an almost compact line. Without loss of generality, let us assume that L2 forms a compact line. The new orientation of the ring follows the smallest distance from the rally line of L2 to the one of L3 (note that, by the definition of rally points, this distance is unique).	
The rally point for FC, call it rp∗, is the middle point of the largest segment containing nodes that are either empty or colored first_color.	
• The agents in FC and da move according to RALLY POINTS CONNECTED FORMATION (Algorithm 5), as follows: (i) agents in FC use rp∗ as rally point; (ii) da uses as rally point the middle point of the almost compact line having its own color.	
• Finally, if after previous point there is only one almost compact line, the two parts of the line (i.e., the dangling agent and all other agents of the line) move towards each other.	

Lemma 6 If the current configuration is correctly placed, then, within finite time, by executing Algorithm 6 (Third Step), ColoredCCP is solved.

Proof. Let us call da the dangling agent of the almost compact line. By definition of Algorithm 6, as long as there is more than one dangling agent (i.e., da and the agents in FC), neither the agents in the compact lines nor those in the almost compact line move. Once the rally point of FC has been computed, the agents in FC and da move according to RALLY POINTS CONNECTED FORMATION, while all others stay still.

By previous Lemmas 3 and 4, within finite time the agents either reach a configuration where ColoredCCP is solved, and the lemma follows, or where there is only one almost compact line and c − 1 compact lines. In the latter case, the only dangling agent and all agents belonging to the almost compact line (note that all of them have the same color) start moving towards each other (according to the smallest distance). Since at most one edge can be missing, within finite time these two parts will meet. Moreover, since rally points computed in Algorithm 4 are distant (2h + 1) from each other, at most two compact lines can overlap. Hence, the lemma follows.

Combining all previous results from this section, we can conclude that:

Theorem 8 Starting from an asymmetric initial configuration, with c ≥ 3 and h ≥ 3, MULTI COLOR CONNECTED SEGMENT algorithm correctly solves the ColoredCCP problem.

Asymmetric initial configuration and h = 2

Now, let us focus on the case where there are c > 2 colors, but there are only two agents for each color (h = 2). In this case, the agents execute the MODIFIED MULTI COLOR CONNECTED SEGMENT algorithm, that follows the lines of MULTI COLOR CONNECTED SEGMENT algorithm, with a minor modification: the agents of the two first colors, say L1 and L2, act as a single group that has the same color. In other words, FC is the union of the agents having the first and the second color in the total ordering of the colors. This change ensures that there are at least three agents in FC, hence the three steps defined by the MULTI COLOR CONNECTED SEGMENT algorithm can still be executed.

Therefore, after the execution of the three steps, agents not in FC form compact lines, while the agents in FC form a segment where agents of two different colors might be interleaved. If the colors of the agents in FC are not interleaved, then ColoredCCP is solved.

Thus, let us assume that the colors of the agents in FC are interleaved: Configuration A in Fig. 11 is, up to symmetries, the only possible configuration. At this point, it is necessary to run a separation procedure that separates the agents of distinct colors, thus forming the remaining two compact lines.

Figure 11 Separating an interleaved line with h = 2 and two colors.

As shown in Fig. 11, from Configuration A, it is possible to reach either Configuration B or Configuration C, by swapping the agents on either edge e1 or edge e3 (at least one of these edges must be available): in both configuration, c − 1 compact lines are formed. At this point, the last two agents (having the same color) have to be compacted: they move towards each other. Note that, since there are at least 2 compact lines of other colors, the configuration remains asymmetric during the movement of these two agents. Thus, since at most one edge at the time can be missing, these two agents will eventually become neighbors, thus solving ColoredCCP. Thus, we just showed the following:

Theorem 9 Starting from an asymmetric initial configuration, with c ≥ 3 and h = 2, the modified version of Modified Multi Color Connected Segment algorithm solves the ColoredCCP problem.

Initial configuration with a mirror symmetry and c > 2

We now consider the last remaining case for ColoredCCP with c > 2 colors: the initial configuration has a mirror symmetry.

Theorem 10 Starting from an initial configuration that has a mirror symmetry (hence, is not periodic), and not compact, the ColoredCCP problem for c > 2 is not solvable if eitherThe axis of symmetry passes through two empty nodes, or,

The axis of symmetry passes through one edge and one empty node, or,

The axis of symmetry passes through two edges and c > 3.

Proof. We prove each of the statements independently.The proof follows directly from Theorem 6.

By hypothesis, the symmetry axis intersects the ring on a node v and an edge e. Therefore, the agents can form the compact lines either around v or around e. If the lines are formed around v, since the ring is not oriented, two agents with the same color would move to v, thus violating the no collision requirement of the problem. If the lines are formed around e, then there would be three intersecting compact lines of three different colors around e, thus violating the ColoredCCP specification.

Since the configuration has a mirror symmetry, the compact lines have to be centred around the symmetry axis. By construction, it is only possible to form two disjoint compact lines. Since there are more than three colors, by the pigeonhole principle, either three compact lines will intersect or there is a pair of intersecting compact lines, thus violating the specification of ColoredCCP.

Note that previous theorem holds for any ring, even non dynamic ones.

Algorithm 7 solves the two remaining cases: (a) the axis of symmetry passes through at least one occupied node; (b) there is an axis of symmetry passing through two edges, and c = 3. We can thus conclude that:

Algorithm 7 MULTI COLOR MIRROR SYMMETRY.

Pre-condition: Initial configuration is aperiodic and with an unique axis of symmetry.	
(a) The axis of symmetry passes through at least one occupied node.	
We follow the statements of Case (b) in Algorithm 2. In particular, since the configuration is not periodic, it is always possible to elect one among the agents that are on the axis of symmetry, let this agent be a. We distinguish the three possible cases:	
1. If the neighbor nodes of a are empty, a moves of one position, and the configuration becomes asymmetric. Now, MULTI COLOR CONNECTED SEGMENT of Section 5.1 can be run.	
2. If the neighbor nodes of a are occupied, and the axis of symmetry passes through another node b, and the neighbor nodes of b are empty, then b moves of one position, and the configuration becomes asymmetric. Now, MULTI COLOR CONNECTED SEGMENT of Section 5.1 can be run.	
3. Finally, no node on the symmetry axis can move. In this case, since the configuration has a symmetry axis, there must be two block of nodes of equal size to the left and to the right of a. These two block of nodes move away from a of one position. Now, either the configuration becomes asymmetric (one of the two block does not move because of a missing edge), or previous Case a.1 applies.	
(b) The axis of symmetry passes through two edges, and c = 3.	
Let e be one of the edges intersected by the symmetry axis, elected as in Case (a) of Algorithm 2. The agents proceed as follows: at each round, only agents with maximum color are allowed to move. In particular, the two agents nearest to e that do not belong to a full segment containing e, move towards e. If no agent is blocked by an edge removal, the symmetry axis is preserved and eventually all agents with maximum color form a full segment around e. Otherwise, if an agent is blocked, the next configuration becomes asymmetric; thus we can apply the Algorithm 1.	
Once we have a compact segment of the first color, following the same strategy, the second color in the order will form a full segment around the antipodal edge e′ of e. Finally, the agents of the third color form a full segment around edge e.	

Theorem 11 If the initial configuration is aperiodic, it has a symmetry axis and either (a) the axis of symmetry passes through at least one occupied node, or (b) there is an axis of symmetry passing through two edges, and c = 3, then Algorithm 7 solves ColoredCCP.

Proof. Let us consider the two possible cases.(a) The axis of symmetry passes through at least one occupied node. First, note that by running Algorithm 7, within finite time the configuration becomes asymmetric. At this time, if h > 2 then algorithm MULTI COLOR CONNECTED SEGMENT (Asymmetric initial configuration and h ≥ 3) can be applied, and the proof follows by Theorem 8. If h = 2, the proof follows by Theorem 9.

(b) There is an axis of symmetry passing through two edges, and c = 3. According to Algorithm 7, agents of different colours are compacted sequentially according to the total ordering of the colors. Let us consider the first color c¯ such that the agents with colors c¯ do not form a full segment. We will first show that, given the edge e elected for compacting the agents around it, with e as defined by Case (b) of Algorithm 7, the agents will indeed form within finite time a compact line centered around e.

Let a and b be two symmetric agents with color c¯ that do not belong yet to the full segment of color c¯ that contains e. If there is no such pair of agent, then segments for all colors are full, and the theorem follows. These two agents move towards e using two different edges, let them be ea and eb. Two possible scenarios may occur. (i) Either ea or eb is missing: in this case only one among a and b moves, leading to an asymmetric configuration. Now, Algorithm 1 can be applied and, by Theorem 5, the theorem follows. (ii) Edges ea and eb are both alive: in this case, after both a and b move, their distance to S decreases. Now, by iterating the argument, either the theorem follows by previous Case (i), or within finite time a and b will eventually join S. In the latter case, the number of symmetric pairs with color c¯ not in the full segment being formed around e decreases. Hence, by induction on all pairs of symmetric agents with color c¯, and on the number of colors, we can conlcude that within finite time the theorem follows.

ColoredCCP with Global Snapshot and c = 2

In the previous section we presented the case for c > 2 colors; here, we analyse the case with c = 2 colors. Let us first notice that the impossibility of Theorem 2 still holds in this case; we thus need to investigate the asymmetric case and the case with a mirror symmetry.

Asymmetric initial configuration

Let us first consider the case of an asymmetric initial configuration.

Theorem 12. If c = 2 and the initial configuration is asymmetric, ColoredCCP is solvable.

Proof. We distinguish two cases:(a) If h = 2 (i.e., four agents in total, two for each color), by Theorem 5, within finite time Algorithm 1 lets the agents form a compact line (where agents of different colors might be interleaved). At this time, the agents can be separated within finite time by using the technique described in Asymmetric initial configuration and h = 2.

(b) If h > 2, let c1 and c2 be the two colors. First note that, since the initial configuration is asymmetric, it is possible to establish a total order among all agents with color c1 (resp., c2). We again distinguish two possible cases. (i) If the agents of color c1 (resp., c2) are placed asymmetrically, they execute Algorithm 1; by Theorem 5, within finite time agents with color c1 (resp., c2) will form a compact line. (ii) Otherwise, the first agent with color c1 (resp., c2) that can move, makes a move that makes the set of all agents with color c1 (resp., c2) to become asymmetric, and previous Case (i) applies.

Hence, within finite time, the theorem follows.

Initial configuration with a mirror symmetry

Let us now consider the case of a symmetric initial configuration. By Theorem 6, it follows that if the initial configuration has a mirror symmetry, is aperiodic, and not compact and the axis of symmetry passes through two empty nodes, then ColoredCCP is not solvable. In the following, we will show that in all other cases the problem is solvable.

Theorem 13 If the initial configuration is aperiodic and has an unique axis of symmetry and either (a) the axis of symmetry passes through at least one occupied node, or (b) there is an axis of symmetry passing through one edge, and c = 2, then ColoredCCP is solvable.

Proof. We distinguish two cases.

(a) The axis of symmetry passes through at least one occupied node. In this case, by running Algorithm 7, within finite time an asymmetric configuration is reached. At this time, theorem follows by Theorem 12.

(b) The axis of symmetry passes through one edge. In this case, by running Algorithm 7, within finite time, two compact lines are formed, and the theorem follows; or the configuration becomes asymmetric, and the theorem follows by Theorem 12.

Conclusions

The study of autonomous agents in distributed networks, and the study of dynamic networks are interesting problems by themselves. Even more interesting is the study of their combination. The results presented in this paper are tight on this track, and despite the simple definition of the problem, its solution hides several difficulties that are strictly related to the changing nature of the underlying network and to the fact that our solutions do not rely on the use of memory of the past (oblivious agents), giving them the nice property of self-stabilization.

In particular, we introduced and studied the Compact Configuration Problem and the Colored Compact Configuration Problem for a set of autonomous mobile agents on a dynamic ring networks. We showed that both problems can be solved only if the initial configuration is aperiodic.

Note that if the agents agree on a common sense of orientation then any aperiodic configuration is asymmetric and thus, in this case the compaction problems can be solved if and only if the initial configuration is not periodic. When the agents do not have a common sense of orientation as in this paper, then we need to also consider those configuration that are mirror symmetric. In such cases, the problems can be solved only under certain conditions.

The results of this paper provides the exact characterization of the solvable initial configurations for the CCP and ColoredCCP problems. We also showed that having persistent memory is not necessary for solving the problem (except in the special case of two agents). It would be interesting to determine what additional capabilities of the agents would allow them to the solve the ColoredCCP problem without any overlaps. Future investigations on this problem could also consider other graph topologies under either the same or a more relaxed model for dynamicity. Another interesting issue is to consider less synchronous models where all agents may not start at the same time and they may not be active at the same time.

There are still few interesting problems that need to be considered in the future:

• When c ≥ 2, we admit the presence of an overlap between at most two lines. When this cannot be avoided?

• When c ≥ 2, we need that the ring is 2hc + c, i.e., to be large enough to not overlap lines when using rally points. What is the lower bound on this quantity? Can we solve the probem in a ring having size hc + c?

• Under which conditions is it still possible to solve the problem when more than one edge might be missing at each round?

• What is the impact of having a semi-synchronous or asynchronous scheduler?

Supplemental Information

Supplemental Information 1 Basic impossibility proof.

Computer-assisted proof for Theorem 1: the code explores the space of all possible solutions.

Click here for additional data file.

A special thanks goes to Linda Pagli, who participated in the first revision of this paper.

Additional Information and Declarations

Competing Interests

Author Contributions

Data Availability

1 A preliminary version of this work appeared in Das et al. (2019b).

The authors declare that they have no competing interests.

Shantanu Das conceived and designed the experiments, performed the experiments, analyzed the data, performed the computation work, authored or reviewed drafts of the paper, and approved the final draft.

Giuseppe Antonio Di Luna conceived and designed the experiments, performed the experiments, analyzed the data, performed the computation work, prepared figures and/or tables, authored or reviewed drafts of the paper, and approved the final draft.

Daniele Mazzei conceived and designed the experiments, analyzed the data, authored or reviewed drafts of the paper, and approved the final draft.

Giuseppe Prencipe conceived and designed the experiments, performed the experiments, analyzed the data, performed the computation work, prepared figures and/or tables, authored or reviewed drafts of the paper, and approved the final draft.

The following information was supplied regarding data availability:

The focus of the paper is on algorithms to control a set of autonomous mobile entities, and on proving their correctness via formal proofs, hence no data are neither involved nor necessary. We are using a computer-assisted method to prove Th. 1, whose code can be either found here: https://colab.research.google.com/drive/1W1H27vdTLC3cEs2rYc2k8TO3ivbppjOR or in the file provided in the Supplemental Material

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
