# Peer review of "Compacting oblivious agents on dynamic rings"

_PeerJ Computer Science, doi:10.7717/peerj-cs.466_

## Round 0.1 · original submission · Minor Revisions

Both reviewers find merit in your work, but they have some minor suggestions to improve the manuscript.

Reviewer 1 ·

Basic reporting

This paper investigates the possibility/impossibility of Compact Configuration Problem (CCP) and Colored Compact Configuration Problem (Colored CCP) of anonymous and oblivious agents in anonymous ring networks. The CCP requires that agents initially deployed arbitrarily should be located in a continuous segment. In the ColoredCCP, each agent has its own color, and agents with the same color should be located in a continuous segment. The paper clarifies the conditions of solvability of these problems.

The results presented in the paper are interesting and valuable sufficiently for publication in PeerJ COmputer Science. The paper is well-organized and easy to read.

The followings are minor comments.

lines 75-81 (mobile agents within dynamic graphs):
The authors should mention that there are two different models concerning the relationship between the adversary and agents. One is that the adversary determines the missing edges first and then agents determine their actions, which is the model considered in this paper). The other is that the adversary determines the missing edges after agents determine their actions.

line 124: "E:N \rightarrow V \times V" should be "E:N \rightarrow 2^{V \times V}".

line 136: Color 1 is considered as max_color but it is confusing since 1 is the minimum among the color set [1..c]. It seems better to use "min_color" instead of "max_color". Also, see the comment to line 420.

line 142: The usage of "blocked" is a little bit confusing for the following reason, and it seems better to make a notice to avoid confusion. Each agent determines its action based on the (global or local) snapshot that indicates whether the edges incident to the current node is missing or not. It seems strange that an agent chooses a missing edge as the direction of movement to be blocked. I guess that "blocked" is introduced for simplicity of algorithm description. If so, it seems better to make a notice.

line 156: "Round" is defined here, but it is used in several places before (lines 123, 124, 127, etc.).

lines 163, 164: It seems better to replace "\mathcal{G}" with "the footprint G" or simply "G". A dynamic graph \mathcal{G} is a sequence of graphs so "v \in \mathcal{G}" is confusing.

lines 188-196 (Proof of Theorem 1): The proof contains some unclear points. (1) I cannot understand what the "states" imply. Are they "configurations"? (2) I cannot understand what configuration the sequence, for example, "000100111", represents. I guess that "0" (resp. "1") represents that the node is empty (resp. occupied by an agent). But we consider here two-colored agents and the information of colors seems missing in the sequence.

around Theorem 1: It is better to say something (or possible/impossible/unknown) for the case of c=2 and h=3.

line 212: The term "disconnected" is undefined. It seems to be replaced with "not compacted". But, it is redundant since any periodic configuration is not compacted.

line 230: It is not clear what local view is represented by [{0,1},{1,1},{1,1}]. I guess that the first, the second, and the last multisets describe the states of nodes at distances 1, 2, and 3 respectively. But the multiset is not appropriate since it cannot distinguish the node in the clockwise direction from that in the counter-clockwise direction.

line 255: The paper says "with only two non-neighbors agents in the ring, the configuration is a palindrome". Does it hold for the configuration [010001]? It is not a palindrome configuration with two agents. (A configuration is a palindrome if some cyclic rotation of this sequence is a palindrome, which is ww^R or wcw^R (where w^R is a reverse of string w, and c is a symbol).

line 290: S_1 and S_2 should be the "maximal" full segments.

Algorithm 1, the definition of S_1 and S_2: S_1 and S_2 should be the "maximal full segments" rather than "non-empty segments".

Algorithm 1, case (a): The direction of movement is not clear from the phrase "all agents not in S_1 move towards S_1". The same for line 317.

line 326: "empt" should be "empty".

Figure 7: It is not a palindrome configuration. Please see the comment to line 255.

Algorithm 2: "neighbors nodes" (three places) should be "neighbor nodes" or "neighboring nodes". The same for Algorithm 7 (three places).

line 355: The paper says "if only one of them exists, then the configuration cannot be palindrome". Does it hold for configuration [110010011]? Here the node of the middle point of the sequence is occupied by agent a. This configuration satisfies the condition of (a) (the axis of symmetry passes through at least one edge), but it is a palindrome. This configuration also satisfies the condition of (b), so it can be considered in (b).

Algorithm 2, (b), 3: "there must be two (full) segments of equal size" should be "there must be two maximal (full) segments of equal size".

line 368: "S^− and S^+ be the two segments" should be "S^− and S^+ be the two maximal full segments"

line 375: "C>2" should be "c>2" (a lower case letter).

lines 392, 393: "with color FC" should be "with color max_color".

line 397: "each color but FC" should be "each color but max_color"

line 402: "a color different from FC" should be "a color different from max_color".

line 419: Here the color set is [0,c-1], but it was defined as [1,c] in Section 2 (line 133).

line 420: "A color c_i" should be "A color i". The same for Definition 4 (two places).

line 420: "c_0=max_color" should be "0=max_color". But it seems strange and the term "max_color" should be "min_color".

Algorithm 4: It is not clear what "the first" implies in the phrase "the first node in FC". I understand we can determine a specific agent in FC and v_f as the node the agent is now located at. It seems better to modify the phrase.

Algorithm 4, Step 1: The paper says "FC is now used to compute c−1 rally points", but actually c (rather than c-1) rally points (including rp_0) are determined using FC (notice that rp_0 is called the first “rally point”). It is confusing.

Lemma 3: The paper says "the system reaches a configuration with c−1 almost compact rally lines". But the configuration can contain compact rally lines. If a compact rally line can be considered as an "almost" compact rally line, the claim holds; otherwise, the claim does not hold. It is not clear from Definition 3.

Algorithm 5, Pattern 1: There are three cases. The first and third both consider the case where there is a missing edge on p. So it seems better to exchange the first and the second cases to improve readability.

Figure 8: It should be explained that the bold circle represents a rally point.

line 465: "thus" should be "Thus".

Lemma 5: When FC forms an almost compact line, then there may exist "two" almost compact lines, which is not correctly placed.

Algorithm 6: It is not clear how to define the distance from L_2 to L_3 when L_3 forms an almost compact line. The distance should be defined from the rally lines of L_2 and L_3.

Algorithm 6: "colored FC" should be "colored max_color".

Subsection 5.2: Section 2 assumes h>2 (line 135) but here the case of h=2 is considered, which is confusing.

line 506: "e_2" should be "e_3".

line 532: The paper says "these three compact lines will intersect". But it is possible that two compact lines will intersect around one of the edges, and another two compact lines will intersect around the other (antipodal) edge. Any case of them violates the requirement of ColoredCCP, so the theorem can be proved.

line 564: "C=2" should be "c=2" (a lower case letter).

line 565: "we presented the case for c = 2 colors; here, we analyse the case with more two colors" should be "we presented the case for c > 2 colors; here, we analyse the case with two colors".

line 567: "case" should be "cases".

Experimental design

no comment

Validity of the findings

no comment

Additional comments

no comment

Reviewer 2 ·

Basic reporting

This paper considers compaction for oblivious agents on dynamic ring networks. This manuscript is almost well written and easy to understand except several parts (stated later). It also has sufficient references for this research and is self-contained.

Experimental design

This research is original one for mobile agents on dynamic rings and combination of mobile agents and dynamic rings is interesting. Also the results presented in the manuscript are almost tight in the sense that the possibility results are given for the cases except the impossibility results for these two problems CCP and CCCP.

Validity of the findings

Although the impact and novelty given to distributed algorithmic field are not so large, the result is useful for researchers working on mobile agents and fault-tolerant ring networks. Conclusions are well stated and contain several interesting open questions.

Additional comments

This paper considers Compact Configuration Problem (CCP) and its colored version (CCCP) on dynamic rings, which changes their topology changes continuously at unpredictable locations and unpredictable times. The authors show the impossibility and possibility results for CCP and CCCP on dynamic rings. That is, the results in this paper provide a full characterization of the solvable instances for both CCP and CCCP.

This manuscript is well written and easy to understand except several parts and so this manuscript can be accepted if the following conditions are satisfied.

(1) Since the relation between the impossibility results and possibility ones is very complicated, some tables which show these relations and clarify the conditions of the impossibility and possibility are necessary and should be inserted.
(2) Since also in the impossibility results there are several cases holding without the assumption of dynamic rings, these are stated explicitly.
(3) The proof of Theorem 3: In Figure 2, a_1-a_8 have the same view. Why the authors differentiate a_3-a_6 and a_1,a_2,a_7,a_8? It should be clarified.
(4) The sentences of the beginning of Section 6 is incorrect. In the previous section, the case for c>2 is presented? Also Algorithm 7 (b) treats the case of c=3 but Theorem 13 shows the result for c=2. Something is wrong. The authors should correct this section.
(5) The following terms should be defined: (local) view and its notation (line290). Although the symmetricity of configuration is defined, but the symmetricity of a set of agents is not defined.
(6) Some typos: line 135, h > 2 should be h \geq 2? The caption of Figure 3 should be Proof of Theorem 4.

---

## Round 0.2 · accepted · Accept

I consider that the manuscript is now suitable for publication.

Reviewer 1 ·

Basic reporting

This paper investigates the possibility/impossibility of the Compact Configuration Problem (CCP) and the Colored Compact Configuration Problem (Colored CCP) of anonymous and oblivious agents in anonymous ring networks. The CCP requires that agents initially deployed arbitrarily should be located in a continuous segment. In the ColoredCCP, each agent has its own color, and agents with the same color should be located in a continuous segment. The paper clarifies the conditions of solvability of these problems.

The results presented in the paper are interesting and valuable sufficiently for publication in PeerJ Computer Science. The paper is revised carefully with considering the previous review comments, and is now can be accepted for publication at PeerJ Computer Science.

Experimental design

The subject of the paper is within Aims and Scope of the journal and attractive to many readers of the journal..

Validity of the findings

The presented results are new. The paper is well written and easy to understand due to the detailed and precise proofs. The conclusion is also well described.

Reviewer 2 ·

Basic reporting

Since the conditions the reviewer asked in the reviews are satisfied, this article is accepted.

Experimental design

There is no comment about the current article.

Validity of the findings

The article satisfies the reviewer's conditions.

---

## Author Rebuttal · Round 0.2

# PeerJ Computer Science

## Compacting Oblivious Agents on Dynamic Rings

Rebuttal letter for Article ID: 55115

We thank the reviewers for their useful comments and suggestions. We reviewed the paper taking all of them into account; a list of the specific actions taken in the revision follows.

## Reviewer 1

- lines 75-81 (mobile agents within dynamic graphs): The authors should mention that there are two different models concerning the relationship between the adversary and agents. One is that the adversary determines the missing edges first and then agents determine their actions, which is the model considered in this paper). The other is that the adversary determines the missing edges after agents determine their actions.

*Done.*

- line 124: "E:N \rightarrow V \times V" should be "E:N \rightarrow 2^{V \times V}".

*Done.*

- line 136: Color 1 is considered as max_color but it is confusing since 1 is the minimum among the color set [1..c]. It seems better to use "min_color" instead of "max_color". Also, see the comment to line 420.

*We agree with the reviewer: we addressed the issue by using "first_color".*

- line 142: The usage of "blocked" is a little bit confusing for the following reason, and it seems better to make a notice to avoid confusion. Each agent determines its action based on the (global or local) snapshot that indicates whether the edges incident to the current node is missing or not. It seems strange that an agent chooses a missing edge as the direction of movement to be blocked. I guess that "blocked" is introduced for simplicity of algorithm description. If so, it seems better to make a notice.

*We added a note to better explain the meaning of "blocked".*

- line 156: "Round" is defined here, but it is used in several places before (lines 123, 124, 127, etc.).

*The reviewer is right: we fixed this.*

- lines 163, 164: It seems better to replace "\mathcal{G}" with "the footprint G" or simply "G". A dynamic graph \mathcal{G} is a sequence of graphs so "v \in \mathcal{G}" is confusing.

*Done.*

- lines 188-196 (Proof of Theorem 1): The proof contains some unclear points. (1) I cannot understand what the "states" imply. Are they "configurations"? (2) I cannot understand what configuration the sequence, for example, "000100111", represents. I guess that "0" (resp. "1") represents that the node is empty (resp. occupied by an agent). But we consider here two-colored agents and the information of colors seems missing in the sequence.

*The reviewer is right, and our method was not properly described. The proof has been reshaped.*

- around Theorem 1: It is better to say something (or possible/impossible/unknown) for the case of c=2 and h=3.

*Done.*

- line 212: The term "disconnected" is undefined. It seems to be replaced with "not compacted". But, it is redundant since any periodic configuration is not compacted.

*Done.*

- line 230: It is not clear what local view is represented by [{0,1},{1,1},{1,1}]. I guess that the first, the second, and the last multisets describe the states of nodes at distances 1, 2, and 3 respectively. But the multiset is not appropriate

since it cannot distinguish the node in the clockwise direction from that in the counter-clockwise direction.

*We addressed this comment, by changing the definition of the local view in the proof of Theorem 3, and by modifying the proof accordingly.*

- line 255: The paper says "with only two non-neighbors agents in the ring, the configuration is a palindrome". Does it hold for the configuration [010001]? It is not a palindrome configuration with two agents. (A configuration is a palindrome if some cyclic rotation of this sequence is a palindrome, which is ww^R or wcw^R (where w^R is a reverse of string w, and c is a symbol).

*The reviewer is right. We thus deleted any reference to "palindrome" throughout the paper, and used mirror symmetry instead.*

- line 290: $S_1$ and $S_2$ should be the "maximal" full segments.

*Done.*

- Algorithm 1, the definition of $S_1$ and $S_2$: $S_1$ and $S_2$ should be the "maximal full segments" rather than "non-empty segments".

*Done.*

- Algorithm 1, case (a): The direction of movement is not clear from the phrase "all agents not in $S_1$ move towards $S_1$". The same for line 317.

*Done.*

- line 326: "empt" should be "empty".

*Done.*

- Figure 7: It is not a palindrome configuration. Please see the comment to line 255.

*Done (as detailed above, "palindrome" is not used anymore).*

- Algorithm 2: "neighbors nodes" (three places) should be "neighbor nodes" or "neighboring nodes". The same for Algorithm 7 (three places).

*Done.*

- line 355: The paper says "if only one of them exists, then the configuration cannot be palindrome". Does it hold for configuration [110010011]? Here the node of the middle point of the sequence is occupied by agent a. This configuration satisfies the condition of (a) (the axis of symmetry passes through at least one edge), but it is a palindrome. This configuration also satisfies the condition of (b), so it can be considered in (b).

*The proof of Theorem has been modified, and deleted any reference to "palindrome" as detailed previously.*

- Algorithm 2, (b), 3: "there must be two (full) segments of equal size" should be "there must be two maximal (full) segments of equal size".

*Done.*

- line 368: "$S^-$ and $S^+$ be the two segments" should be "$S^-$ and $S^+$ be the two maximal full segments"

*Done.*

- line 375: "C>2" should be "c>2" (a lower case letter).

*Done.*

- lines 392, 393: "with color FC" should be "with color max_color".

*Done.*

- line 397: "each color but FC" should be "each color but max_color"

*Done.*

- line 402: "a color different from FC" should be "a color different from max_color".

*Done.*

- line 419: Here the color set is [0,c-1], but it was defined as [1,c] in Section 2 (line 133).

*We followed the reviewer's suggestion, and everything has been defined starting from 0.*

- line 420: "A color c_i" should be "A color i". The same for Definition 4 (two places).

*Done.*

- line 420: "c_0=max_color" should be "0=max_color". But it seems strange and the term "max_color" should be "min_color".

*Done.*

- Algorithm 4: It is not clear what "the first" implies in the phrase "the first node in FC". I understand we can determine a specific agent in FC and v_f as the node the agent is now located at. It seems better to modify the phrase.

*We modified the description of the algorithm.*

- Algorithm 4, Step 1: The paper says "FC is now used to compute c−1 rally points", but actually c (rather than c-1) rally points (including rp_0) are determined using FC (notice that rp_0 is called the first "rally point"). It is confusing.

*We clarified the description of the algorithm.*

- Lemma 3: The paper says "the system reaches a configuration with c−1 almost compact rally lines". But the configuration can contain compact rally lines. If a compact rally line can be considered as an "almost" compact rally

line, the claim holds; otherwise, the claim does not hold. It is not clear from Definition 3.

*The statement has been changed, according to the reviewer's suggestion.*

- Algorithm 5, Pattern 1: There are three cases. The first and third both consider the case where there is a missing edge on p. So it seems better to exchange the first and the second cases to improve readability.

*Done.*

- Figure 8: It should be explained that the bold circle represents a rally point.

*Done (in the updated version of the paper, it is Figure 9).*

- line 465: "thus" should be "Thus".

*Done.*

- Lemma 5: When FC forms an almost compact line, then there may exist "two" almost compact lines, which is not correctly placed.

*The definition of "correctly placed" has been changed: the new definition does not consider the agents in FC.*

- Algorithm 6: It is not clear how to define the distance from L_2 to L_3 when L_3 forms an almost compact line. The distance should be defined from the rally lines of L_2 and L_3.

*Done.*

- Algorithm 6: "colored FC" should be "colored max_color".

*Done (changed to the newly introduced "first_color").*

- Subsection 5.2: Section 2 assumes h>2 (line 135) but here the case of h=2 is considered, which is confusing.

*We updated Section 2, specifying h≥ 2.*

- line 506: "e_2" should be "e_3".

*Done.*

- line 532: The paper says "these three compact lines will intersect". But it is possible that two compact lines will intersect around one of the edges, and another two compact lines will intersect around the other (antipodal) edge. Any case of them violates the requirement of ColoredCCP, so the theorem can be proved.

*We updated Theorem 10: we added one more case to the proof following the suggestion of the reviewer.*

- line 564: "C=2" should be "c=2" (a lower case letter).

*Done.*

- line 565: "we presented the case for c = 2 colors; here, we analyse the case with more two colors" should be "we presented the case for c > 2 colors; here, we analyse the case with two colors".

*The introduction of Section 6 has been fixed as suggested.*

- line 567: "case" should be "cases".

*Done.*

# Reviewer 2

- (1) Since the relation between the impossibility results and possibility ones is very complicated, some tables which show these relations and clarify the conditions of the impossibility and possibility are necessary and should be inserted.

*We thank the reviewer for the suggestion: we inserted Table1 and Table2, to summarize the results.*

- (2) Since also in the impossibility results there are several cases holding without the assumption of dynamic rings, these are stated explicitly.

*We thank the reviewer for this suggestion, that actually strengthens our results: we added explicit notes for all proofs that do not need assumption of dynamic rings.*

- (3) The proof of Theorem 3: In Figure 2, a_1-a_8 have the same view. Why the authors differentiate a_3-a_6 and a_1,a_2,a_7,a_8? It should be clarified.

*We better described the local views of the agents.*

- (4) The sentences of the beginning of Section 6 is incorrect. In the previous section, the case for c>2 is presented? Also Algorithm 7 (b) treats the case of c=3 but Theorem 13 shows the result for c=2. Something is wrong. The authors should correct this section.

*The introduction of Section 6 has been fixed as suggested.*

- (5.1) The following terms should be defined: (local) view and its notation (line290).

*We defined the local views of the agents (Theorem 3).*

- (5.2) Although the symmetricity of configuration is defined, but the symmetricity of a set of agents is not defined.

*We updated subsection "Configurations and other definitions" in Section 2. In particular, we introduced the definition of mirror symmetry (eliminating tany reference to "palindrome" that we had in the first version of the paper), and added a new figure.*

- (6) Some typos: line 135, h > 2 should be h \geq 2? The caption of Figure 3 should be Proof of Theorem 4.

*Done.*